# The Insulin-Degrading Enzyme from Structure to Allosteric Modulation: New Perspectives for Drug Design

**DOI:** 10.3390/biom13101492

**Published:** 2023-10-07

**Authors:** Grazia Raffaella Tundo, Giuseppe Grasso, Marco Persico, Oleh Tkachuk, Francesco Bellia, Alessio Bocedi, Stefano Marini, Mariacristina Parravano, Grazia Graziani, Caterina Fattorusso, Diego Sbardella

**Affiliations:** 1Department of Clinical Science and Traslational Medicine, University of Rome Tor Vergata, Via Della Ricerca Scientifica 1, 00133 Rome, Italy; grazia.raffaella.tundo@uniroma2.it (G.R.T.);; 2Department of Chemical Sciences, University of Catania, Viale Andrea Doria 6, 95125 Catania, Italy; grassog@unict.it; 3Department of Pharmacy, University of Naples “Federico II”, Via D. Montesano 49, 80131 Napoli, Italy; marco.persico@unina.it (M.P.); oleh.tkachuk@unina.it (O.T.);; 4Institute of Crystallography, CNR, Via Paolo Gaifami 18, 95126 Catania, Italy; 5Department of Chemical Sciences and Technologies, University of Rome Tor Vergata, Via della Ricerca Scientifica 1, 00133 Rome, Italy; 6IRCCS—Fondazione Bietti, Rome, Italy; mariacristina.parravano@fondazionebietti.it; 7Department of Systems Medicine, University of Rome Tor Vergata, Via Montpellier 1, 00133 Rome, Italy; graziani@uniroma2.it

**Keywords:** insulin-degrading enzyme, Alzheimer’s disease, diabetes, proteasome, insulin, allostery

## Abstract

The insulin-degrading enzyme (IDE) is a Zn^2+^ peptidase originally discovered as the main enzyme involved in the degradation of insulin and other amyloidogenic peptides, such as the β-amyloid (Aβ) peptide. Therefore, a role for the IDE in the cure of diabetes and Alzheimer’s disease (AD) has been long envisaged. Anyway, its role in degrading amyloidogenic proteins remains not clearly defined and, more recently, novel non-proteolytic functions of the IDE have been proposed. From a structural point of view, the IDE presents an atypical clamshell structure, underscoring unique enigmatic enzymological properties. A better understanding of the structure–function relationship may contribute to solving some existing paradoxes of IDE biology and, in light of its multifunctional activity, might lead to novel therapeutic approaches.

## 1. Introduction

The insulin-degrading enzyme (IDE), also named insulysin, is a 110-kDa neutral zinc and thiol-dependent metallopeptidase and a member of the invercinzin family of metalloendopeptidases, which are characterized by the inversion of the active site motif “HxxEH” compared to the classical sequence “HExxH”, which is usually found in other metalloenzymes [1,2]. The IDE comprises four structurally homologous αβ domains (domain 1, residues 1–285; domain 2, residues 286–515; domain 3, residues 542–768; and domain 4, residues 769–1019). The N-terminal portion of the IDE (i.e., IDE-N) that includes domains 1 and 2 (colored white and blue, respectively; Figure 1) and the C-terminal portion (i.e., IDE-C) that includes domains 3 and 4 (colored pink and magenta, respectively; Figure 1). IDE-N and IDE-C are joined by a 28-residue hinge loop (colored red; Figure 1).

In accordance with its name, the IDE was discovered for its capability to cleave insulin (see Section 1.1) and for its primary role in the regulation of insulin metabolism [3,4].

Nevertheless, the IDE is an evolutionary ancient enzyme, and its homologs have been identified in phylogenetically distant organisms, spanning from eukaryotes to bacteria. Surprisingly, the primary sequence of the IDE displays a high similarity across kingdoms, envisaging the existence of enzymatic functions conserved through evolution [5,6].

In humans, the IDE is encoded by a gene mapped on chromosome 10 and is synthesized as a single polypeptide [2,7,8].

Both the transcript and the protein of the IDE show a ubiquitous expression in human tissues and a primary cytosolic localization. However, the IDE has been described in several subcellular compartments, including endosomes, peroxisomes, and mitochondria, and is associated with plasma membrane, albeit signal sequences targeting the enzyme to these cellular compartments have not been characterized [9,10,11].

Moreover, although this occurrence is still debated, about 1–10% of the total IDE is thought to be secreted into the extracellular milieu through an unconventional pathway, as its secretion rate is unaffected by canonical inhibitors of the endoplasmic reticulum to the Golgi pathway, such as brefeldin A, in cell cultures [12,13,14,15,16,17].

The documented presence of the IDE in evolutionarily distant organisms, and the multiple subcellular localizations discussed above, which often do not overlap with its presumed natural substrates, support the growing hypothesis that the biological role of the enzyme is much broader than originally thought [18,19].

The IDE degrades a wide spectrum of peptides with little or no sequence homology, including β-amyloid (see Section 3.3), amylin, glucagon, somatostatin, insulin-like growth factor-2 (IGF-2), dynorphins, bradykinin, and atrial natriuretic peptides, suggesting a potential role in patho-physiological processes modulated by these peptides [20,21,22,23]. Moreover, during the last years, several studies have pointed out that the real IDE biological function may take place through non-catalytic activities [24,25].

Further, when broadening the spectrum of molecular mechanisms described for the IDE, it has been proposed that the enzyme interacts with the varicella zoster virus (VZV) glycoprotein E and plays an important role in VZV infection and virus cell-to-cell spread [26,27,28,29].

Notably, several studies have underscored a key role of the IDE in the regulation of proteostasis and a close structural link with the proteasome (see Section 2.3) [22,30].

Therefore, before IDE targeting may truthfully become a therapeutic option for one out of the different human diseases, dysregulation of IDE expression or activity has been associated with several structural/functional topics, and the biological features of the enzyme must be definitively addressed.

In this framework, this manuscript provides an update of the state of the art about the biochemical properties, structural–functional relations, and patho-physiological roles of the IDE within the pathogenesis of human diseases.

### 1.1. The Discovery of the IDE and Its Role in Insulin Catabolism

In 1949, Mirsky and Broh–Kahn identified a mixture of proteases showing the ability to degrade insulin in rat tissues [31]. In the following years, a single enzyme that cleaves insulin, but not proinsulin, was purified from rat skeletal muscles [3,32]. Then, a bulk of studies have confirmed [33,34,35,36] that the “insulin protease” was implicated in the proteolysis of the hormone in extracts of muscle, liver, kidney, pancreas, fibroblasts, and red blood cells [32,35,37,38,39]. The cleavage of insulin A and B chains, which compose the mature hormone, led to the IDE being characterized. The A chain is cut at residues A13–A14 and A14–A15, whereas the cleavage of the B chain takes place in correspondence with the B9–B10, B10–B11, B13–B14, B14–B15, B16–B17, B24–B25, and B25–B26 peptide bonds [40,41,42]. The relative abundance of the detected insulin fragments varies, depending on the experimental conditions [43]. Remarkably, Duckworth and co-workers highlighted that IDE cleavage at B16-B17 residues could alter insulin interaction with its receptor since B16 tyrosine is a crucial residue for insulin–receptor recognition [40]. In support of an IDE role in insulin clearance, it was reported that (i) insulin peptide fragments isolated by cells are consistent with insulin peptides released by hormone degradation by the IDE in vitro and (ii) insulin internalization and degradation were altered by non-specific IDE inhibitors (i.e., N-ethylmaleimide and bacitracin) and upon delivery of anti-IDE antibodies in living cells [40,41,44,45,46]. An intriguing point has concerned the investigation into the possible cellular sites of insulin degradation by the IDE. A two-model process has been proposed.

First, the degradation of receptor-bound insulin by a membrane-associated IDE before its internalization has been envisaged. Conversely, studies in hepatocytes have supported a different hypothesis. In these cells, a large amount of receptor-bound insulin was reported to be shunted to endosomes, where the complex dissociated and insulin was degraded by endosomal proteases, such as Cathepsin D, neutral aminopeptidase, and the IDE [46,47,48,49].

The hypothesis that the IDE could also degrade the two separate insulin chains must also be taken into account, and different insulin fragments could be produced in this case, some of which may have important biological activities differing from the ones of the full-length hormone [50].

In this context, the previously cited analogy between insulin B chain fragments generated in cells and in vitro by the IDE envisaged that the endosomal compartment was actually the intracellular localization where the IDE biological role took place. In contradiction with this possibility, even moderate acid pH values, such as those of endosomes, were found to inactivate the IDE [51]. To better circumstantiate this issue, it has been then postulated that the IDE mediates the primary step of hormone degradation in early endosomes when the hormone is still bound to its receptor, and pH still falls within a physiological range [41,46,48,52,53,54,55,56].

Nowadays, the physiological relevance of insulin degradation by the IDE in vivo is largely questioned, as recently extensively reviewed elsewhere [57]. Briefly, if some studies using different transgenic models have reported that insulin level in plasma significantly increases in the absence of the enzyme (IDE-KO mice), several other studies have reported unchanged insulin levels following genomic knock-out of the enzyme. Additionally, it has been reported that mice harboring liver-specific knock-out of the IDE gene display normal insulin levels in the case of a regular diet and elevated levels when fed a high-fed diet (see Section 2.1) [57,58,59,60,61,62].

## 2. Pills on IDE Multiple Roles in Health and Disease

Different aspects of IDE biology, spanning from its conservation through evolution, its broad subcellular localization, the variety of peptides that it degrades, and its recently proposed non-proteolytic functions [2,19,22,63], prompted us to consider the IDE as a moonlight enzyme, a term referred to proteins performing different and often unrelated functions acquired through evolution, with multiple roles in health and diseases [2]. A bulk of studies, extensively reviewed elsewhere, have indicated the IDE as a potential target in two main therapeutic areas; i.e., metabolic and neurodegenerative diseases, even though, recently, it has been proposed that its pharmacomodulation could be an innovative therapeutic approach to treat cancer [22]. In this paragraph, we briefly summarize the main evidence of IDE involvement in the pathogenesis of type 2 diabetes mellitus (T2DM) and Alzheimer’s disease (AD), and in the regulation of intracellular proteostasis.

### 2.1. The IDE and Type 2 Diabetes Mellitus (T2DM)

T2DM is a syndrome characterized by the gross impairment of metabolism and hyperglycemia caused by a low rate of insulin production/secretion by β-cells of Langheran’s islets in the pancreas and/or an alteration in hormone signaling. Patients with diabetes have an increased risk of developing severe diabetes-specific complications such as neuropathy, nephropathy, and retinopathy [64]. From the beginning, Mirsky, who discovered the IDE and its role in insulin catabolism (see Section 1.1), reasoned that its inhibitors would have offered a new antidiabetic therapy [65].

Over the past decades, IDE’s role in the onset and progression of T2DM has been long envisaged, and it has been recently reviewed [41]. Genetic polymorphisms within the IDE locus have been linked with an increased risk of T2DM in different ethnicities [66]. Under physiological conditions, IDE expression is under the control of insulin signaling. Hence, insulin resistance results in low IDE levels and, as a consequence, in slower insulin turnover. This behavior seems to lead to a vicious cycle that sustains hyperinsulinemia, contributing to the desensitization of the insulin receptor in peripheral tissues and T2DM progression [67,68]. However, studies on IDE-KO mice are controversial since some models show pronounced glucose intolerance and hyperinsulinemia, whereas others are not hyper insulinemic [69]. Moreover, in humans, the IDE level is significantly reduced in muscles and the liver but increased in the serum of T2D2M patients with respect to healthy individuals [6,66,70]. It appears evident that the involvement of the IDE in T2DM pathogenesis is still ambiguous and apparently depends on a range of factors that influence the specific study design. In this context, it is not surprising that the pharmacological inhibition of IDE activity has produced contradictory outcomes (see Section 3.1). Treatment of lean and obese mice with the selective IDE inhibitor (3*R*,6*S*,9*S*,12*E*,16*S*)-9-(4-Aminobutyl)-3-[(4-benzoylphenyl)methyl]-6-(cyclohexylmethyl)-2,5,8,11,14-pentaoxo-1,4,7,10,15pentaazacycloeicos-12-ene-16-carboxamide (6bK) was found to induce a decrease in postprandial glucose concentrations after oral glucose administration (see Section 4.1 [71]). On the other hand, another IDE inhibitor, the methyl *N*-[(2S)-2-[4-[5-[4-[[(2*S*)-2-[(3*S*)-3-amino-2-oxopiperidin-1-yl]-2-cyclohexylacetyl]amino]phenyl]pentoxy]phenyl]-3-quinolin-3-ylpropyl]carbamate (NTE-1), was reported to not affect insulin degradation in vivo, even though an improvement of glucose excursion in oral glucose tolerance test in this case was reported [72]. This different behavior could be explained by different pharmacokinetics, as well as the pharmacodynamics of 6bK and NTE-1. However, further studies are needed to shed light on the association between IDE inhibition and the dynamics of glycemic metabolism.

As mentioned, some data reported a reduced IDE level in T2DM. Therefore, one problem concerns the effect of a further decrease in its activity by the administration of specific inhibitors. In fact, it can be speculated that treatment with IDE inhibitors may not be beneficial, especially in view of the additional biological roles played by the enzyme [57,73,74].

Furthermore, chronic hyperinsulinemia induced by IDE inhibition may result in increased insulin resistance over time. Nevertheless, the IDE degrades many metabolically active substrates, including glucagon and amylin. As a matter of fact, after the administration of 6bK to T2DM mice, glucagon and amylin levels were increased in the intraperitoneal glucose tolerance test. As a consequence, IDE inhibitors could cause an imbalance in the insulin–glucagon ratio and amyloid deposition. Interestingly, several studies revealed an increased risk of developing AD in T2DM patients [73]. The evidence of an insulin signaling defect in AD individuals has prompted researchers to consider AD as “a type 3 diabetes”. Since the IDE covers the double role of insulin and Aβ degrading enzyme, a possible role as a linker of the two pathologies has been proposed [2,74,75]. Moreover, the effects on the whole-body IDE deletion should be deeply investigated (see Section 2.2 and Section 2.3).

### 2.2. The IDE and Alzheimer’s Disease

The IDE shows a broad expression into the central nervous system (CNS) [76,77], suggesting a role in brain functionality. Although conflicting data are reported, IDE expression is reduced during brain aging. This seems to be associated with a dysregulation of insulin metabolism and the accumulation of some amyloidogenic proteins, which are typically found also in healthy-aged individuals [63,78,79]. In recent years, the IDE has attracted attention as a potential therapeutic target for AD, mainly due to its role in β-amyloid (Aβ) clearance [13,66]. AD is the most prevalent CNS degenerative disease and is characterized by progressively cognitive impairment, which causes a reduction in daily living. The pathogenesis of AD is multifactorial, and the archetypal phenotypic signature is the accumulation of Aβ and hyperphosphorylated tau protein in amyloid plaques in selected neuroanatomical areas of the brain [80,81,82]. The intraneuronal accumulation of toxic oligomeric forms of Aβ is believed to be the primary event that causes synaptic damage and neuronal death [19,81]. The alteration of Aβ clearance may result in the progressive accumulation of aggregation-prone toxic species (the “amyloid” cascade hypothesis) [83]. Several proteases are known to degrade Aβ, including neprilysin and the IDE [84]. The IDE is the main enzyme involved in Aβ clearance in the cytosol of human brain lysates and in the cerebrospinal fluids [19,85,86,87]. Moreover, it has been reported that the IDE regulates Aβ level in vivo and factors that alter IDE expression and activity can lead to an increase in this level [18,23,88]. Interestingly, the IDE seems to decrease in the early AD phases, whereas it is overexpressed after the development of the first Aβ deposits.

The IDE colocalizes at the periphery of Aβ plaques in the cerebral cortex of AD transgenic mice, where it is supposed to be exposed to oxidation and a reduction in its activity [87,89,90,91,92]. In accordance with this possibility, the IDE has been reported to be more oxidized in the AD hippocampus than in the cerebellum of the AD transgenic model and in AD patients [89,93]. This pathological modification of the enzyme is likely part of a vicious cycle that further leads to Aβ accumulation and aggregation, as well as an increase in oxidative stress [19,91]. Accordingly, in an AD brain, high levels of an S-nitrosylated inactive IDE were reported [94,95,96,97]. Another important aspect of bridging the IDE with AD is the positive correlation between some genetic variants of the enzyme and the risk of developing late-onset AD, even though, in this case, some controversies still exist [7,98,99,100,101]. As mentioned in Section 2.1, a number of studies, as recently reviewed, reveal an increased risk of developing AD in T2DM patients, and the two diseases share common features, spanning from the alteration of insulin signaling and glucose metabolism to insulin resistance and inflammation [102,103,104,105]. However, the nature of the relationship has remained undefined. Anyway, the IDE is considered a key factor in the crosstalk between T2DM and AD, and its dual role in degrading insulin and β-amyloid should be taken into consideration in therapeutic strategies targeting the IDE for the management of these two pathologies [2,75].

#### Involvement of the Insulin-Degrading Enzyme in Retina Pathology

Concerning the role of the IDE in the CNS, recent evidence underlines the IDE’s role in retina physiopathology. The retina is a full-fledged part of the CNS that converts the electrical activity of photoreceptors into action potentials that travel to the brain via axons in the optic nerve. It shares multiple patho-physiological features with the brain. As a matter of fact, retina neurodegenerative processes are linked to an irreversible loss of neurons, which culminate in retinal dysfunction. Recently, it has been proposed a novel function for the IDE in the physiopathology of retinitis pigmentosa, a group of rare genetic dystrophies thatare associated with primary dysfunction and death of rod photoreceptors and a loss of cones, leading to constriction of the visual field and eventually blindness. Currently, retinitis pigmentosa is neither preventable nor curable [106,107]. Interestingly, mammalian cone inner segments reveal high levels of the IDE in physiological conditions, whereas the IDE is downregulated in the dystrophic retina of mouse models of retinitis pigmentosa carrying distinct mutations. Additionally, treatment of the retinal degeneration rd10 mouse model of retinitis pigmentosa with a synthetic peptide analog of the preimplantation factor, which is a natural peptide with neuroprotective functions that positively modulate IDE expression and activity, preserved retinal structure and delayed loss of visual function [106,108,109]. These studies clearly open a novel perspective in the biology of the IDE and its therapeutic potential.

### 2.3. The IDE as a Playmaker in the Regulation of Proteostasis

The involvement of the IDE in the regulation of proteostasis is a challenging but promising area of research on the biology of this enzyme. In this context, the activities/properties described so far through which the IDE is expected to contribute to protein homeostasis are referred to (i) “dead-end” chaperone; (ii) Heat Shock Protein (HSP); and (iii) interaction with components of the Ubiquitin Proteasome System (UPS) (Figure 1) [2]. With respect to the first point, in addition to its ability to degrade amyloidogenic proteins, it has been proposed that the IDE non-proteolytically interferes with the aggregation of amyloidogenic peptides, such as Aβ and α-synuclein. In fact, the IDE exosite (see Section 3.2) was reported to entrap the Aβ monomer and the C-terminal domain of α-synuclein forming stable non-catalytic complexes [110,111,112,113]. Furthermore, trapping the Aβ monomer seems to prevent its oligomerization, reducing the rate of seeding formation [110,113]. With respect to the aggregation of these peptides, although they are unlikely to fit into the IDE catalytic chamber, the overall folding of these macromolecular structures has been envisaged to be remodeled by the enzyme [114]. However, the molecular mechanisms underscoring this property have not been characterized. Recently, the chaperone activity of the IDE toward Aβ has been better characterized using an inactive IDE mutant (IDE_E111Q_) (see Section 3.1). In this case, Aβ and the IDE seem to form a complex in which the substrate can be entrapped without being cleaved [115]. As a whole, the double skills through which the IDE interacts with Aβ are suggested to prevent toxic oligomer formation [63]. The second IDE function that reinforces its role in the response to unfolded protein accumulation is the behavior of HSP. In particular, it was demonstrated that the IDE level increases after different stressful conditions and is required for neuronal cell survival in vitro through a pattern similar to HSP70. Furthermore, IDE downregulation (by delivery of antisense oligonucleotides) inhibits cell proliferation and viability probably acting on the UPS since it leads to a decrease in the overall content of poly-ubiquitinated proteins [116,117]. The UPS is the main pathway involved in the turnover of intracellular soluble proteins tagged with ubiquitin polymers, thus serving key roles for cell life [118,119]. The key end-point of this pathway is the 26S proteasome, a multi-catalytic assembly composed of the 20S proteasome core particle, which houses the proteolytic activity, capped by one (the “single-capped” 26S) or two 19S (the “doubly-capped” 30S) regulatory particle(s) [118,119]. The IDE has been proposed (i) to interact and modulate the activity of the 20S proteasome catalytic core and (ii) to control the equilibrium between the different proteasome particles, 20S, 26S, and 30S, which physiologically populate the cell cytosol in vitro. Therefore, even though the physiological relevance of this interaction must be unveiled, it has been proposed that the IDE binding to the 20S induces a conformational change in the proteasome structure that modulates its catalytic activity and prevents it from being capped by the 19S, increasing the pool of uncapped 20S, at least in vitro [25,30,116]. The 26S is known to target substrates conjugated with ubiquitin polymers. The ubiquitin conjugation proceeds through a three-step process, involving, in the first step, the ubiquitin-activating enzyme E1, which activates the ubiquitin monomer in an ATP-dependent manner, and, in the second and third steps, an ubiquitin-conjugating enzyme E2 and an ubiquitin (E3) ligase, respectively, that, sequentially, mediate the conjugation of activated ubiquitin to target substrates [120]. Although controversies still exist, it is worth recalling that the IDE has been proposed to exert functions in the ubiquitination machinery process. In fact, the IDE has been proposed as a non-canonical ubiquitin-activating enzyme. In the absence of E1 but in the presence of E2 enzymes, it promotes the formation of ubiquitin dimers [24]. The physiological meaning of this novel role remains unclear, but it is worth exploring to cast definitive light on the contribution of the IDE to UPS functioning.

## 3. The “Atypical Structure” of the IDE

Several X-ray crystallography and cryo-electron microscopy studies have been undertaken to solve the structure of the IDE and its correlation with the functional properties underscoring the enigmatic enzymological mechanisms of the IDE [41]. Thus, a better understanding of the structure–function relationship may help solve some existing paradoxes of IDE biology, including the biomolecular mechanisms involved in catalytic and extra-catalytic activities.

### 3.1. The IDE Structure and Biochemical Properties Relationship

Firstly, the structure of the monomeric human IDE was resolved by Shen and colleagues [121]. Nowadays, it is known that the IDE is composed of four homologous αβ domains: domains 1 and 2 (colored white and blue, respectively; Figure 1) form the N-terminal portion of the IDE (i.e., IDE-N), whereas domains 3 and 4 (colored pink and magenta, respectively; Figure 2) form the C-terminal one (i.e., IDE-C). The former and the latter are joined by a 28-residue hinge loop (colored red; Figure 2). Indeed, the overall structure of the IDE resembles a clamshell, where IDE-N and IDE-C are bowl-shaped halves. The extensive hydrogen bond between them creates a “latch” [2,4,20,121] (Figure 3). Notably, mutations that destabilize the hydrogen bond latch improve catalytic efficiency [122]. They define an inner chamber that allows the binding and the degradation of the substrates [85,121]. The catalytic chamber of the IDE can entrap substrates with a size of approximately 6 kDa (see Section 3.3). The catalytic activity is located in the IDE-N, whereas IDE-C has a key role in substrate recognition as well as IDE oligomerization (see below) [123]. Domain 1 contains the conserved “HxxEH” zinc biding motif (see Section 1), where the catalytic metal ion is coordinated by His108, His112, and Glu189 residues (Figure 2). In addition to the catalytic triad, the Glu111 residue serves crucial roles in the catalytic cycle by mediating the activation of a catalytic water molecule (Figure 2). Accordingly, the replacement of this residue (E111Q) with Gln (E111Q residue) renders the IDE almost completely catalytically inert [113,121,124]. Interestingly, the entrapment of the Aβ monomer has been investigated in the chamber of E111Q. The latter was found to inhibit amyloid aggregation, suggesting that the proposed “dead-end chaperone” function of the IDE (see Section 2.3) is independent of its catalytic activity [113]. Recently, it was reported that when the *wt* IDE entraps a stable random coil/α-helix Aβ dimer, the IDE does not prevent the contacts between the monomers, and it contributes to the inhibition of Aβ aggregation of these stable dimers by its “dead-end” chaperone-like activity. When the IDE entraps a less stable random coil/α-helix Aβ dimer, the IDE can impede the contacts between the monomers. Although the IDE does not fully encapsulate these unstable dimers, it prevents some of the contacts between the monomer, inhibiting the nucleation phase [125]. With its 140 residues, α-synuclein is unlikely to be an IDE substrate; nonetheless, similar to what has been reported for Aβ, the IDE forms SDS-resistant complexes with α-synuclein in vitro and efficiently inhibits α-synuclein aggregation in a non-proteolytic manner [111]. Although this is a very fascinating topic, the structural basis and the molecular mechanism underscoring the dead-end chaperone activity of the IDE is largely unknown, and it deserves more attention mainly for the rational design of drugs that could inhibit Aβ and α-synuclein aggregation. The overall structure of the IDE resembles a clamshell, where IDE-N and IDE-C are bowl-shaped halves. The extensive hydrogen bond between them creates a “latch” that maintains the enzyme in a “closed” inactive conformation [2,4,20]. Notably, mutations that destabilize the hydrogen bond latch improve catalytic efficiency [122]. During the catalytic cycle, the IDE adopts at least two major conformational states: the open-state IDE, which captures substrates and releases products, and the closed-state IDE, which is associated with catalysis [126]. Thus, a transition between the “closed” and “open” states is required to make the active site accessible to substrates and represents the rate-limiting step of the IDE catalytic mechanism [127]. Importantly, a “swigging door” mechanism allows the IDE to capture substrates of different sizes. A smaller conformational transition, which should occur more frequently, might be sufficient to capture and degrade smaller peptides, whereas larger substrates would only be able to enter the catalytic chamber of the IDE following a wider opening (see Section 3.3) [126,128]. Understanding the structural basis of the open–closed transition should contribute to unveiling not only the mechanism through which the IDE captures its substrates and releases the reaction products, but it might also provide insights into how the IDE facilitates the unfolding of its substrates prior to catalysis, as well as how the IDE carries out its non-proteolytic activities [74,111,126].

In the solution, the IDE exists as a mixture of monomers, dimers, and tetramers, and the dimer has been proposed as the predominant and more active species (Figure 4). The dimer interface is formed by elements of the two C-terminal domains of the IDE monomer (domain 3 and 4, colored pink and magenta, respectively; Figure 4), including a sheet-like contact formed by a terminal β-strand [129,130]. The key role played by the C-terminus in inter-subunit communication is demonstrated by data, indicating that the removal of a sequence of 18 residues from the C-terminus stabilizes IDE monomers [128,130]. Thus, modulating the open–closed transition, as well as the equilibrium between the oligomeric forms of the IDE, are two suitable strategies to affect IDE activity [2] (see Section 3.3).

During the catalytic cycle, the IDE adopts at least two major conformational states: the open-state IDE (Figure 4A), which captures substrates and releases products, and the closed-state IDE, which is associated with catalysis [126] (Figure 4C). In the dimeric form, the transition between the open and the closed states is characterized by the presence of an intermediate semi-open conformation, as evidenced by Cryo-EM studies [126] (Figure 4B). The transition between the “closed” and “open” states is required to make the active site accessible to substrates and represents the rate-limiting step of the IDE catalytic mechanism [127]. Importantly, a “swigging door” mechanism allows the IDE to capture substrates of different sizes. A smaller conformational transition, which should occur more frequently, should be sufficient to capture and degrade smaller peptides, whereas larger substrates would only be able to enter the catalytic chamber of the IDE following a wider opening (see Section 3.3) [126,128]. Understanding the structural basis of the open-closed transition should contribute to unveiling not only the mechanism through which the IDE captures its substrates and releases the reaction products, but it should also provide insights into how the IDE facilitates the unfolding of its substrates prior to catalysis, as well as how the IDE carries out its non-proteolytic activities [74,112,126].

### 3.2. The Heterogeneity of IDE Substrates

As reported in the previous sections (see Section 1), insulin has been the first identified IDE substrate [48] (Figure 5A and Figure 6). However, the IDE cleaves a number of substrates spanning from Aβ, amylin, glucagon, and somatostatin to neuropeptides in vitro [2,21,131,132] (Figure 1). The IDE substrates can be didactically divided into two groups: (i) the macromolecular substrates that exceed 12–15 amino acids in length, such as amylin, Aβ, insulin, and glucagon (Figure 5A), and (ii) short peptides (β-endorphin, kinins, somatostatin). The IDE substrates share little sequence homology, and substrate recognition is proposed to rely upon their tertiary and quaternary structures rather than their primary sequence. Anyway, the IDE shows cleavage preferences for basic (i.e., arginine and lysine) and/or hydrophobic residues (i.e., phenylalanine, leucine, and tyrosine) located at P1′ of the target proteins [53]. A distinguishing feature shared by macromolecular IDE substrates is the tendency to unfold and form aggregates under certain chemical-physical circumstances [19]. Intriguingly, it has been proposed that IDE substrates may lose their proper conformation before being accommodated into the catalytic chamber where β-sheet structural interactions with the β6 strand (residues S137−S143) of the IDE active site (Figure 5, colored red) are supposed to take place [133]. In support of these mechanisms, IDE residues A140, F141, and Y150, which are contained on the β6 strand, are critical for IDE–substrate interaction [133]. Furthermore, the size of the IDE catalytic chamber clearly indicates that only monomers can be entrapped and degraded. The structural model for the catalytic cycle indicates that the C-terminal tail of substrates binds to the catalytic site (Figure 5, see Section 3.1), and the C-terminal charge distribution affects substrate affinity since peptides with positive charge residues are characterized by low affinity and poor degradation [121,134]. Larger macromolecular substrates are instead known to bind their N-terminal tail to an additional site, called exosite, located in domain 2 about 30 Å off the active site [135], whereas the N-terminus of short peptides does not interact with this site, and the recognition depends only on the catalytic site [136,137]. It is reasonable that the great variety of substrates could reflect different interaction modes and catalytic specificities (Figure 5 and Figure 7A). In this respect, peptides with fewer positive charges like atrial natriuretic peptide, glucagon, and IGF-2 easily avoid the repulsive forces of IDE-C and are better substrates than positively charged ones, such as brain natriuretic peptide, insulin-like growth factor-1 IGF-1, and glucagon-like peptide 1. Moreover, in general, short peptides reveal a reduced substrate affinity for the IDE, which is counterbalanced by a faster rate-limiting step for catalysis [63,121,138,139]. Concerning the two most studied IDE macromolecular substrates, Aβ and insulin, it is important to recall that the catalytic efficiency toward the former is higher than that calculated for the latter [2,135]. Moreover, unlike insulin, Aβ degradation displays a slight cooperativity, suggesting that the binding of one molecule of Aβ to a subunit of the IDE dimer facilitates the degradation of a second one by the other IDE subunit [135]. A specific mention deserves ubiquitin since some studies have shown that ubiquitin could be a substrate of the IDE [140]. However, these data remain unclear and not easily reproducible [141].

### 3.3. The IDE as an Allosteric Enzyme

Two structural features of the IDE are crucial in the regulation of its activity: (i) the existence of one (or multiple) exosite(s) topologically distinct from the active site and (ii) the presence of a ligand-linked equilibrium between monomers (Figure 2), dimers (Figure 4), and tetramers (Figure 6) [6,114]. As a matter of fact, two modes of allosteric modulators are described for the IDE, namely homotropic and heterotrophic [2]. The first type of functional modulation envisages the binding of homotropic molecules to the catalytic site of one subunit in the dimeric and/or tetrameric assembly that affects the activity of the adjacent one. Conversely, the heterotrophic allosteric modulator binds to a pocket distinct from the catalytic site and affects the catalysis within the same subunit [142,143]. The exosite (see Section 3.1 and Section 3.2) binds macromolecular substrates to the N-terminus of the IDE, thus permitting the cleavage cycle at the catalytic site. Interestingly, some short substrates, such as kinins and endorphins, which do not require binding to the exosite for cleavage (see Section 3.2), can exclusively bind to the exosite with higher affinity than the catalytic center. Accordingly, the X-ray structure of the complex IDE-bradykinin revealed the binding of bradykinin to the exosite and not to the catalytic site (Figure 6A) [85]. This binding increases the rate of the hydrolysis of another molecule of a short substrate (but not a macromolecular substrate) at the catalytic site [85,144]. However, the short substrate somatostatin has been proposed to affect IDE activity on macromolecular substrate (i.e., insulin and Aβ) binding at least two heterotrophic sites, one coinciding with the canonically identified exosite [21,23,129,145]. The IDE has been also shown to be modulated heterotrophically by polyphosphate anions, in particular by ATP, other nucleotides, and triphosphate moieties [129,146,147,148] that activate the IDE toward a short peptide. The binding sites for ATP (Figure 7B) largely overlap with those proposed for somatostatin [2,129,145]. It suggests that the same IDE topological area can bind to different classes of heterotrophic modulators (e.g., small ligands and anions), even though the functional effects should vary, depending on the specific modulator and the substrate investigated [2,129,145,149]. For the hydrolysis of insulin and Aβ, somatostatin has a stimulatory effect at sub-micromolar concentration, whereas ATP inhibits it [145,146]. At higher concentrations, somatostatin acts as a negative regulator of Aβ processing. It has been proposed that this effect could be related to a homotropic conformational change in the IDE [2,21,135]. On the other hand, studies of IDE activity in mixed dimers, in which only one subunit displays mutation affecting its activity, reveal that the modulation by small peptides occurs only in the subunit where the ligand is bound (“cis” effect) [150]. Additionally, a stable monomeric IDE mutant, lacking the IDE-C portion responsible for dimerization (IDE^Δ*C*^) (see Section 3.1), loses the ATP activating effect, suggesting that ATP can also act on the quaternary structure of the enzyme [128,130,151,152]. These and other observations underline the possibility that a heterotopic ligand could also affect the activity of the partner subunit, acting as a homotropic modulator [2]. The structural basis for homotropic modulation is proposed to be connected to the structural arrangements of the dimer, in which one subunit is in a closed conformation, whereas the partner one is in a partially open state. The binding of modulators to one subunit induces a conformational rearrangement that should be transmitted to the adjacent one [126,128]. Detailed information concerning this transition is lacking and represents the main goal of future structural–functional relationship studies [2].

#### Metal Ions Affect IDE Activity

Another class of IDE activity modulators is represented by metal ions [153]. In this case, a dramatic influence on the enzyme activity has been observed using copper ions [153,154,155]. Particularly, our group has found that while Cu^2+^ is capable of inhibiting the enzyme activity in a reversible manner (activity is restored after Zn^2+^ addition), Cu^+^, as well as Ag^+^, are strong and irreversible IDE inhibitors. Indeed, unbiased MD simulations predict that Cu^+^ blocks the two IDE cysteines, Cys-812 and Cys-819, directly altering the cavity size by more than 15 Å and leading to a dramatic structural perturbation. The latter causes the locking of the catalytic site, which, therefore, becomes inaccessible to host the substrates. On the contrary, copper(II), having a higher affinity than zinc(II) for the imidazole nitrogen atoms of peptide ligands that form macrochelate rings, is capable of substituting the Zn^2+^ present at the catalytic site, bringing about the inhibition of enzyme activity due to the different coordination requirements of the two metal ions. However, such inhibition is reversible, because when the amount of added zinc(II) reaches a value that overcomes the different affinity, zinc(II) substitutes copper(II) in the catalytic site, and IDE activity is restored [154,155].

## 4. Pharmacological Modulation of the IDE: A Therapeutic Perspective

As already reported in the previous sections, the IDE is considered a therapeutic target for different pathologies. So far, two different therapeutic approaches have been taken into consideration, namely inhibition and activation of its catalytic activity, which are considered promising for T2DM and AD, respectively (see Section 4.1 and Section 4.2) [66,149]. This type of intervention is conceptually linked to the IDE degrading activity on insulin and β-amyloid and the involvement of the enzyme in the onset and progression of these pathologies (see Section 2). Anyway, as discussed, controversial data on the efficiency of IDE modulation, mainly in the case of T2DM, are reported (see Section 2.1). Many aspects of IDE’s real role in cell life are also unclear, and this makes the design of appropriate therapeutic strategies based on IDE modulation very difficult. The IDE is able to degrade a variety of different substrates, and it is shown to have multiple functions that are not fully understood. This poses a unique challenge for drug development because of an increased potential for off-target side effects [136]. An important aspect that is underestimated is the effect of enzyme modulators on the non-proteolytic functions of the IDE, which deserves more attention. In fact, it appears vital for the developing of a suitable therapeutic approach to investigate the effect on the UPS system and on “dead end chaperone” activity [63]. In the next sections, the two main approaches, namely IDE inhibition and activation, are discussed separately.

### 4.1. The IDE Inhibitors

The design of the IDE substrate-selective inhibitors, which can affect degrading activity only toward a specific target, thus reducing, at least in principle, the side effects generated by the degradation of other substrates [66], has been proven a very challenging approach. As reported in Section 3, it is important to recall that small molecules that bind to the IDE exosite, leaving the catalytic site free, could alter substrate selectivity [71]. Additional viable strategies are the selective modulation of the extracellular IDE or the identification of allosteric modulators of IDE activity [129,151,156], even though it is unclear how these strategies should encompass the side effects due to the high number of IDE substrates. Until now, two main different classes of IDE inhibitors have been identified. The first one targets the zinc catalytic site, while the second one targets the IDE exosite. The former approach has been widely investigated, and several peptide hydroxamic acid-based inhibitors have been proposed [157] (Figure 8). Although in principle targeting the zinc catalytic site could hinder IDE selectivity over other zinc metalloproteases, Leissring and coworkers clearly showed that it is possible to develop inhibitors that cross-react minimally with conventional zinc metalloproteases thanks to the distinctive structure of IDE’s active site (see Section 3.1). Such work produced the highly selective li1 inhibitor, with IC_50_ ≈ 0.6 nM, which is much more potent than the most common zinc metalloprotease inhibitors, such as 1,10 phenanthroline (IC_50_ ≈ 300 µM) or BDM44768 (IC_50_ ≈ 60 nM). Moreover, it was also reported that thanks to a subsequent characterization of a diverse set of conventional and retro-inverso peptide hydroxamates, it is also possible to achieve a discrete substrate specificity, a highly unexpected finding for active site-directed inhibitors [158]. However, substrate-dependent activity modulation has been observed in other cases [159]. For example, ATP inhibits the degradation of insulin and Aβ (see Section 3) but activates the degradation of other short fluorogenic substrates because its binding was shown to occur at a site distinct from the active site (Figure 7B vs. Figure 5A); thus, it acts as a noncompetitive activator [148]. Further studies reveal that polyanions, such as ATP, can shift the oligomeric state of the enzyme from dimer-tetramers to a monomer, which explains why they are activators only toward small substrates. Moreover, a binding site for polyanions has been suggested to exist on the same extended surface that forms one wall of the substrate-binding chamber. Interestingly, in the case of longer substrates, ATP causes charge-induced structural modifications in the active site, where a simultaneous interaction between ATP, a long substrate, and the IDE occurs. Such interaction exists only when both ATP and the long substrate are simultaneously present in the catalytic chamber, resulting in an allosteric, noncompetitive inhibition with an apparent decrease in substrate affinity [151].

Although IDE inhibitors targeting the exosite have been developed only recently (Figure 9), many of them have been already well-characterized (bacitracin, IC_50_ ≈ 300 µM; NTE-1, IC_50_ ≈ 4 nM; etc.) [160]. The most comprehensive screening of IDE inhibitor candidates has been carried out by Maianti and coworkers [161]. After performing an in vitro selection on a DNA-templated library of 13,824 synthetic macrocycles for their ability to bind an immobilized mouse IDE, they came up with a shortlist of 6 candidate IDE-binding molecules, which, eventually, led to the identification of the inhibitor 6bK (IC_50_ = 50 nM) as the best-performing IDE inhibitor. Indeed, the most valuable finding of Maianti’s study was that the selectivity of 6bK in vitro was ≥1000-fold for the inhibition of the IDE over all other metalloproteases tested, a very promising result for the in vivo applications. It is important to highlight that 6bK should bind the IDE at the exosite binding pocket according to the experimentally determined structure of the human IDE in a complex with its analog 6b (Figure 9). Such an inhibition mechanism is the reason for the high specificity of 6bK for the IDE with respect to other metalloproteases since this exosite is not present in the latter [71,136]. Finally, it is worth mentioning that other IDE inhibitors targeting IDE cysteine residues have also been developed in an attempt to selectively target the extracellular IDE, which is present in an oxidized environment, while sparing the intracellular IDE within the reduced environment of the cytosol. For this purpose, ML34 has been developed and characterized. It has an IC50 for the IDE of about 20 nM [156]. Anyway, many issues still need to be considered if such strong IDE inhibitors must be efficiently used in vivo. First, where the actual in vivo IDE inhibition would take place. Second, the correct evaluation of the systemic effect of IDE inhibition. In order to answer these questions, it is necessary to continue in vivo investigations for a better understanding of the mechanism of inhibition, so that the design of more substrate-specific inhibitors with valid therapeutical applicability will finally become feasible [162].

For the reasons outlined above, designing and optimizing a drug that could be effective in treating neurodegenerative diseases through IDE activity modulation is a very challenging task. Recently, Azam and co-authors have carefully reviewed the possibility of designing and using IDE inhibitors as a valid alternative to treat diabetes rather than using injected exogenous insulin [163]. In this perspective, the ideal IDE inhibitor to treat diabetes should inhibit the clearance of insulin and amylin without affecting the catabolism of other IDE substrates, such as glucagon or amyloid beta peptides. Specificity can be achieved often if the inhibitor binds to both the N-terminus anchoring exosite and catalytic site of the IDE. Indeed, for example, the lower binding affinity of Aβ for IDE (~100-fold lower than the one for insulin) is postulated to contribute to the preferential inhibition of BDM41367 to specifically suppress the degradation of Aβ and not insulin [74]. For the same reason, IDE inhibitors acting as allosteric regulators, binding away from the catalytic cleft, can be designed to differentially affect the levels of selected IDE substrates. The same considerations can also be performed in the case of IDE activators; although, in this case, it is also possible to take advantage of possible synergistic activation effects, as observed, for example, for Ia1 or Ia2 tested in the presence of ATP [149], which is discussed in the next paragraph.

### 4.2. The IDE Activators

If several strategies and/or small molecules have been proposed and investigated to inhibit IDE activity in pathologies, such as diabetes, the activation of the IDE has been targeted to tackle pathological conditions where the accumulation of misfolded proteins represents the prominent biomolecular feature. In this scenario, using a synthetic reporter based on yeast a-factor mating pheromone precursor cleavable by multiple IDE orthologs, many small molecules stimulating rat IDE activity in vitro were found. However, when the same molecules were tested on the human IDE and Aβ-based reporters, the results were much less convincing [164]. Therefore, although it is clear that it is possible of activating the IDE in a substrate-specific manner, finding a suitable molecule for certain specific aims proved to be very challenging. For example, many natural substances have also been tested for IDE activity-positive modulation and, as many of these are capable of crossing the blood–brain barrier, they could be good candidates for therapeutical purposes [159]. Polyphenols such as resveratrol (Figure 10) have been reported to affect the cryptic peptides generated by the action of the IDE on Aβ but not insulin, demonstrating once again the possibility of activating the IDE in a substrate-specific manner [165]. In this perspective, although resolving the molecular features involved in the IDE activation mechanism by the small molecules can be puzzling, it is nonetheless possible to study the effect that the modulators have on the enzyme cooperativity by indirect measurements. Indeed, the IDE has been proven to be a cooperative enzyme, and the Hill coefficient has been estimated to be about 1.8 [150]. Although the experimental techniques normally applied to test enzyme properties and allosteric regulations depend on the ability to measure the enzymatic activity, the immobilization of the IDE on the surface plasmon resonance gold chip has allowed us to obtain activities measurements [166] but also to have an estimate of the Hill coefficient for the binding of insulin to the enzyme in the presence of different positive and negative modulators [159]. Particularly, it has been demonstrated that the Hill coefficient values range from 0.5 in the presence of a strong inhibitor, such as EDTA, to 3.4 in the case of carnosine (Figure 10). A dipeptide reported to be a specific activator of the IDE toward long (insulin, Aβ) but not short substrates (small fluorogenic synthetic substrates) [159]. Indeed, the IDE seems to play a pivotal role in carnosine neuroprotection from cell Aβ toxicity, but therapeutic perspectives are hindered because of the carnosine bioavailability, and the dipeptide is quickly degraded in the blood stream by carnosinase. An analogous problem of in vivo unavailability has been encountered in the case of suramin, a molecule that is non-permeable across the cell membrane and toxic, and, therefore, not suitable for in vivo studies [167]. Another class of molecules, which are inhibitors of amyloid degradation and activators of the hydrolysis of insulin and IGF-2, as well as capable of binding the IDE to the catalytic site and exosite, have been also proposed [168] (Figure 11). However, none of these compounds have been used for further in vivo investigations, while more effective molecules were screened in an attempt to establish a more solid structure–activity relationship. Indeed, previously proposed molecules, such as suramin, were revealed to be non-permeable across the cell membrane, toxic, and not suitable for in vivo studies [169]. Indeed, more than 8000 potential indole-based activators of the IDE were screened, and the best compound was shown to be the activator of both Aβ and insulin hydrolysis by the IDE and displayed cellular activity in a glucose-stimulated insulin secretion assay. Structure–activity relationships for this indole activators series highlighted the importance of the sulfonylation of the nitrogen of the indole, as sulfonamide and imidazole are key for binding with the enzyme [169]. These findings indicate that the modulation of IDE activity is a very promising approach for therapeutic purposes, as well as a very challenging task due to the complex biomolecular mechanisms involved in enzyme modulation. The research aimed at finding the right molecule capable of fulfilling all the in vivo requirements for IDE activity modulation is still an open and challenging field.

## 5. Concluding Remarks

In sum, although early studies underlined that the IDE plays a crucial role in insulin metabolism, its pharmacological inhibition as a strategy to treat diabetes has cast considerable doubts. The role of the IDE in degrading amyloidogenic proteins also remains not clearly defined. Moreover, the novel non-proteolytic functions of the IDE that have been recently proposed render more intricate solving of the biology and pharmacology of the IDE. In light of IDE’s multifunctional activity with strong links to different human diseases, expanding our understanding of IDE biology might lead to novel therapeutic approaches.

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
