# Peer review of "The Insulin-Degrading Enzyme from Structure to Allosteric Modulation: New Perspectives for Drug Design"

_biomolecules, 2023, doi:10.3390/biom13101492_

Round 1
Reviewer 1 Report
The main scope of this review is a good overview over the Insulin Degrading Enzyme, an important protein of many functions. The manuscript is a complete review over the IDE role and potential pharmacological target. This is a nice and solid paper, and definitely worth publication.
There is no doubt that the topic and article are original and relevant to the field of not only enzymology but also neuroscience, the manuscript includes text describing further prospects for the application of various pharmacological tools, such as design of inhibitors and activators of this enzyme. thus this review has a broader scientific meaning, also for pharmacology and drug design.
This article is a very solid review of all aspects of the Insulin Degrading Enzyme, its multiple roles in living organisms. The reference list is appropriate and, to my opinion, complete as it includes the most important publications. I don't recall such a complete review in the literature during recent years.
To my opinion, this article is very well written and no further improvements would be necessary, besides minor language corrections. e.g. line 20 (is: "sisease", should be "disease")
Minor corrections of typing errors needed.
Author Response
We are grateful to the reviewer for his/her positive revision of our manuscript.
Reviewer 2 Report
Insulin degrading enzyme is a Zn2+ peptidase originally discovered as the main enzyme involved in the degradation of insulin and other amyloidogenic peptides, such as β-amyloid peptide. But its role in degrading amyloidogenic proteins remains not clearly defined and, more recently, novel non proteolytic functions of IDE have been proposed. From the structural point of view, IDE presents an atypical clamshells structure, underscoring unique enigmatic enzymological properties. A better understanding of the structure-function relationship may contribute to solve some existing paradoxes on IDE biology and, in the light of its multifunctional activity, might lead to novel therapeutic approaches. This manuscript will help to design IDE modulators to combate viarieties of diseases. But I have several Following concerns:
1. Abbreviations should be defined when they first appear, either in the abstract or in the main text.
2. Authors should add a diagram of IDE's various domains and actual amino acid residue positions, as well as a diagram of IDE enzyme substrates and their corresponding functions in the Section "Introduction", which can help readers understand the structure and function of IDE more quickly and effectively.
3. Authors would like to add some famous IDE inhibitors and their IC50 values mentioned in the article to help readers understand the types of common IDE inhibitors and provide reference for the design of IDE targeted drugs.
4. Please show the software used to make the protein molecule interaction picture, the software copyright and the key amino acid residues of IDE and its inhibitor in realated Figures.
5. Please use the flow chart to show the process of drug design and optimization according to the report, and discuss the structure activity relationships of the key drugs and their targets.
6. Please unify the format of references in the article, including the author's name, the case of words in the title of the article, the writing of the name of the journal, and the page number.
Moderate editing of English language required.
Author Response
Insulin degrading enzyme is a Zn2+ peptidase originally discovered as the main enzyme involved in the degradation of insulin and other amyloidogenic peptides, such as β-amyloid peptide. But its role in degrading amyloidogenic proteins remains not clearly defined and, more recently, novel non proteolytic functions of IDE have been proposed. From the structural point of view, IDE presents an atypical clamshells structure, underscoring unique enigmatic enzymological properties. A better understanding of the structure-function relationship may contribute to solve some existing paradoxes on IDE biology and, in the light of its multifunctional activity, might lead to novel therapeutic approaches. This manuscript will help to design IDE modulators to combate viarieties of diseases. But I have several Following concerns:
- Abbreviations should be defined when they first appear, either in the abstract or in the main text.
According to reviewer’s suggestion, the abbreviations have been indicated in the abstract and in the main text
- Authors should add a diagram of IDE's various domains and actual amino acid residue positions, as well as a diagram of IDE enzyme substrates and their corresponding functions in the Section "Introduction", which can help readers understand the structure and function of IDE more quickly and effectively.
R: According to reviewer’s suggestion, a figure was added reporting human IDE sequence, secondary structure, domains and amino acid positions and diagram of IDE enzyme substrates in the Section "Introduction".
- Authors would like to add some famous IDE inhibitors and their IC50 values mentioned in the article to help readers understand the types of common IDE inhibitors and provide reference for the design of IDE targeted drugs.
According to the Reviewer’s suggestions, we have added some text and other relevant references:
Pag. 16: “…Such work produced the highly selective li1 inhibitor, with IC50 ≈ 0.6 nM, which is much more potent than the most common zinc metalloprotease inhibitors such as 1,10 phenan-throline (IC50 ≈ 300 µM) or BDM44768 (IC50 ≈ 60 nM).”
Pag. 17: “…Although IDE inhibitors targeting the exosite have been developed only recently (Figure 8), many of them have been already well characterized (bacitracin, IC50 ≈ 300 µM; NTE-1, IC50 ≈ 4 nM; etc.). (ref. ChemMedChem 2021, 16, 1776 – 1788).
Pag. 18: “…Finally, it is worth mentioning that other IDE inhibitors targeting IDE cysteine residues have also been developed, in the attempt to selectively target extracellular IDE, which is present in an oxidized environment, while sparing intracellular IDE within the reduced environment of the cytosol. For this purpose, ML34 has been developed and characterized, having an IC50 for IDE of about 20 nM. (Ref. Abdul-Hay, S.O.; ACS Chem. Biol. 2015, 10, 2716–2724).
Pag. 18: ref. “Catalytic site inhibition of insulin-degrading enzyme by a small molecule induces glucose intolerance in mice, nature communication 2015” was also added.
- Please show the software used to make the protein molecule interaction picture, the software copyright
R: The software used to make the pictures is properly cited in the figure captions.
and the key amino acid residues of IDE and its inhibitor in related Figures.
R: a 2D diagram showing the protein-inhibitor interactions was inserted in all IDE-inhibitor figures.
- Please use the flow chart to show the process of drug design and optimization according to the report, and discuss the structure activity relationships of the key drugs and their targets.
We appreciated the Reviewer’s comment as it prompted us to better refine our work. Particularly, we came across with a published work by Azam et al., which we had previously missed to cite and discuss. In this work, there is already an updated scheme of key drugs and their targets for IDE, so we have avoided to produce an analogous scheme in our work, but we have rather cited and discussed the Azam’s paper, together with other works already cited in the previous version, in order to better elucidate the strategies applied for the design of IDE modulators, as well as their mechanism of action, as advised by the Reviewer. We have therefore introduced in our revised version the following:
“…For the reasons outlined above, designing and optimizing a drug, which could be effective to treat neurodegenerative diseases through IDE activity modulation is a very challenging task. Recently, Azam et al. (Pharmaceutical Research (2022) 39:611–629) have carefully reviewed the possibility to design and use IDE inhibitors as a valid alternative to treat diabetes rather than using injected exogenous insulin. In this perspective, the ideal IDE inhibitor to treat diabetes should inhibit the clearance of insulin and amylin without affecting the catabolism of other IDE substrates, such as glucagon or amyloid beta peptides. Specificity can be achieved often if the inhibitor binds to both the N-terminus anchoring exosite and catalytic site of IDE. Indeed, for example, the lower binding affinity of Aβ for IDE (~100-fold lower than the one for insulin) is postulated to contribute to the preferential inhibition of BDM41367 to specifically suppress the degradation of Aβ and not of insulin (Tang, Trends Endocrinol Metab. 2016;27(1):24–34). For the same reason, IDE inhibitors acting as allosteric regulators, binding away from the catalytic cleft, can be designed to differentially affect the levels of selected IDE substrates. The same considerations can be done also in the case of IDE activators, although, in this case, it is also possible to take advantage of possible synergistic activation effects, as observed, for example, for Ia1 or Ia2 tested in the presence of ATP (PLoS One. 2009; 4(4): e5274. Already cited), discussed in the next paragraph…”
- Please unify the format of references in the article, including the author's name, the case of words in the title of the article, the writing of the name of the journal, and the page number.
The reference list have been edited
In addition, an error in the reference 175 has been detected and corrected through the manuscript.
Round 2
Reviewer 2 Report
The authors have addressed all my concerns. I recommend accecepting this manuscript.